# Fluorescent Marker as a Tool to Improve Strategies to Control Contaminated Surfaces and Decrease Danger of Cross-Contamination in Dental Clinics, during and beyond the COVID-19 Pandemic

**DOI:** 10.3390/ijerph20065229

**Published:** 2023-03-22

**Authors:** Eran Dolev, Ilana Eli, Ester Mashkit, Naftali Grinberg, Alona Emodi-Perlman

**Affiliations:** Department of Oral Rehabilitation, The Maurice and Gabriela Goldschleger School of Dental Medicine, Sackler Faculty of Medicine, Tel Aviv University, Tel Aviv 6139001, Israel; eran@drdolev.com (E.D.); elilana@tauex.tau.ac.il (I.E.); estermashkit@mail.tau.ac.il (E.M.); naftalig3@gmail.com (N.G.)

**Keywords:** fluorescent marker, infection control, SARS-CoV-2, COVID-19, dental clinics

## Abstract

The COVID-19 pandemic posed an increased threat to dental personnel and patients. Close encounters with patients’ breath and saliva and the use of intraoral rotating instruments which disperse microscopic airborne particles both increase the possibility of environmental infection. In this study, fluorescent marker (FM) was used to assess and enhance surface cleanliness in the dental clinics and public areas of a major dental school. Initially, 574 surfaces in various areas of a dental school were marked with FM for 3 consecutive months to monitor the surface cleanliness. The initial evaluation results were visually presented to both students and para-dental and cleaning personnel during a designated educational session, and were used to stress the importance of preventing cross-contamination. Following educational intervention, 662 surfaces were re-examined for an additional 3 months, using the same method. A significant improvement in the surfaces’ cleanliness (ANOVA, F_(1)_ = 10.89, *p* < 0.005) was observed post-intervention. The results were more prominent in students’ clinics, which were the students’ cleaning responsibility. The results show that fluorescent markers can serve as an educational tool to improve strategies to control contaminated surfaces in large clinics, such as dental schools. Their use can substantially decrease the hazard of cross-contamination during the pandemic and beyond.

## 1. Introduction

The worldwide state of emergency declared in March 2020, and the rapid spread of the COVID-19 pandemic, increased the need to develop protocols to prevent cross-contamination in medical settings, as well as in other public areas [1]. Traumatic events such as the COVID-19 pandemic can have negative physical and psychological consequences, not only among the general public but also among medical and dental personnel [2,3,4]. The situation for dentists and para-dental personnel was especially complex. Economic worries, a fear of contagion, concerns for family and friends, and conflicts concerning patients’ treatment caused considerable psychological distress [2].

Coronavirus disease 2019 (COVID-19) is a novel severe respiratory syndrome caused by a new beta-coronavirus, known as severe acute respiratory syndrome coronavirus 2 (SARS-CoV-2) [5,6,7]. SARS-CoV-2 can be transmitted through droplets, aerosols, or contact (direct or indirect via fomites). Direct transmission can occur through close contact with infected people through secretions (such as saliva and respiratory secretions) or their respiratory droplets which are expelled when a person is in close contact with an infected person who has respiratory symptoms (e.g., coughing or sneezing) or who is talking or singing [8]. Asadi et al. [9] showed that speech has a greater ability to transmit respiratory infectious diseases compared to breathing due to a greater quantity of particles that are emitted when compared to breathing and the larger size of particles which can carry a larger number of pathogens. The authors conclude that physiological factors, varying among individuals, can affect the probability of respiratory infectious disease transmission.

Indirect contact transmission involving contact between a susceptible host and a contaminated object or surface (fomite transmission) is also possible. The SARS-CoV-2 virus can be found on surfaces for periods ranging from hours to days, depending on the ambient environment and the type of surface. Although there are no specific reports which have directly demonstrated fomite transmission for SARS-CoV-2, it is considered a likely mode of transmission, given the consistent findings on environmental contamination in the vicinity of infected cases and the fact that other coronaviruses and respiratory viruses can be transmitted this way [8].

The danger of being infected by the SARS-CoV-2 virus was especially significant for dental personnel [10]. It is now evident that short-range (conversational) and long-range aerosol transmission plays at least some part in how all respiratory viruses are transmitted between people [11]. In a dental office, verbal communication with patients is carried out in close vicinity to the patient’s mouth. Infected individuals could transmit significant numbers of respiratory pathogens via speech, even in the absence of overt clinical signs of illness [9].

Additionally, due to the nature of dental treatments, saliva-contaminated aerosols are routinely created during treatment. It was found that the viral load of SARS-CoV-2 ribonucleic acid (RNA) in participants’ naso-pharyngeal swabs is positively correlated with the RNA viral load that is emitted in both droplets >10 µm and in bio aerosols <10 µm [12]. Respiratory droplets from infected individuals can also land on objects, creating fomites. As environmental contamination has been documented by many reports, it is likely that people can also be infected by touching these surfaces and touching their eyes, nose, or mouth before cleaning their hands [8]. Thus, although COVID-19 fomite transmission has not been demonstrated, until the risk of COVID-19 fomite transmission is fully understood, continued efforts to frequently clean and disinfect environmental surfaces are needed [13].

The control of contaminated surfaces in medical settings is important at all times, but its importance increased during the pandemic. The pandemic emphasized the importance of continuously cleaning, disinfecting, monitoring, and controlling surfaces and equipment [14,15], and increased the need for simple detection tools that would enable continuous safe dental care during the pandemic, as well as in everyday dental practice [16,17].

In 2003, the Center of Disease Control (CDC) updated the guidelines [18] for infection control in dental healthcare, which were adopted by most dental schools around the world [19]. The guidelines recommend maintaining hand hygiene, disinfecting and sterilizing medical equipment, and improving the quality of water in treatment [20]. General cleaning and disinfection with chemical or physical agents (such as ethylene oxide, peroxide hydrogen, and peracetic acid) can reduce the risk of contamination associated with medical equipment [20]. Fortunately, although the SARS-CoV-2 virus may persist on surfaces such as glass, metal, or plastic for up to 9 days, it is very sensitive to the action of disinfectants [21].

Most healthcare facilities use disinfectants approved by the Environmental Protection Agency (EPA), such as sodium hypochlorite solution [22]. According to various tests, the virus responds better to ethanol than isopropanol, following contact of at least 30 s [1]. Tan et al. [23] developed a systematic procedure to establish the correlation between particulate matter (PM), a mixture of solid particles and liquid droplets found in the air, and microbial counts in hospital operating rooms. The procedures verified the operating parameters of air change rate, room differential pressure, relative humidity, and air temperature. The authors proposed the frequent monitoring of PM5 and PM10 before each surgical procedure.

Dental procedures are usually ambulatory and relatively short in nature. Monitoring PM before each procedure is impractical. Nevertheless, the control of contaminated surfaces is crucial to decrease the dangers of cross-contamination. The CDC suggested several modes to test the quality of surface cleanliness. One such tool is a fluorescent gel, which has the advantage of easy use and is useful in detecting spreading infections from bodily fluids [24,25]. A fluorophore is a fluorescent chemical compound that can re-emit light upon light excitation. When a fluorophore absorbs light, its electrons become excited and move from a resting state to a maximal energy level, called the excited electronic singlet state. Fluorescent markers are useful in a wide range of applications, such as in identifying and quantifying distinct populations of cells, cell surface receptors, etc. [26]. Fluorescent tracer dye intervention can also be used to quantify dermal exposure to agrochemicals among farmers and serve as an educational intervention, which reduces the risk of adverse health effects [27].

Different stages of the dental curriculum require students to demonstrate and adopt different skill sets. In essence, the first stage of the curriculum is mostly theoretical. When students progress to the clinical stage of the curriculum, they learn not only new fine motor and precision skills but also contamination control. All dental schools operate within large dental clinics, in which senior dental students practice their skills as future dentists. Preventing cross-contamination through surfaces and educating students to detect and prevent possible contamination became primary goals during the COVID-19 pandemic.

The surfaces most frequently touched inside the dental clinic are drawer knobs, light handles, unit switches, dental radiograph equipment, reusable containers of dental materials, drawer handles, and dental chairside computers [28]. Certain sites such as the handle of the dentist’s chair and computer surfaces are mostly problematic [20]. In the public areas, the most frequently touched surfaces are door handles, countertops, waiting room chairs, elevator buttons, etc. [15,29].

The aim of this study was to assess the cleanliness of surfaces in clinics and in the public areas of a major dental school and to evaluate the effectiveness of fluorescent markers as an educational intervention to control contaminated surfaces and decrease the dangers of cross-contamination.

## 2. Materials and Methods

The study received the approval of the academic head and the chief administrative director of the dental school, and was carried out at the dental clinics and public areas of the School of Dental Medicine, Tel Aviv University, Israel. It was carried out during the first year of the COVID-19 pandemic (June 2020–May 2021).

### 2.1. Study Design (Figure 1)

Pre-intervention evaluations (pre-I): A total of 13 evaluations (on different days) of various surfaces’ cleanliness were carried out, twice a week, during a period of 3 months, as described below. Students, faculty members, administrative staff, and the managers and employees of an external cleaning company responsible for cleaning the public areas were oblivious to this part of the study. Personal protective equipment (PPE) was mandatory for all personnel. Students’ PPE included gowns, medical protective masks, goggles, gloves, disposable cap masks, disposable clothing, and full-face holds. Cleaning personnel used disposable clothing, medical protective masks, and gloves.

Intervention: There was an intervention process during which the initial findings were displayed and demonstrated to students, faculty, administrative staff, and managers of the cleaning company who take care of the public areas. Each of the relevant groups was invited to an educational session in which pre-I results were visually presented and the importance of cleaning was explained and emphasized. An updated cleaning protocol, based on the CDC 2003 and CDC COVID-19 recommendations [18,30], was formulated and presented. Stations of disinfectant solution (chlorhexidine gluconate 0.5% and ethanol 70%) were scattered throughout the relevant areas and subjects were encouraged to use them at all times.

Post-intervention evaluations (post-I): A total of 9 evaluations (at different days) were carried out twice a week 6 months after the educational intervention, with the use of FM, as described below.

**Figure 1 ijerph-20-05229-f001:**
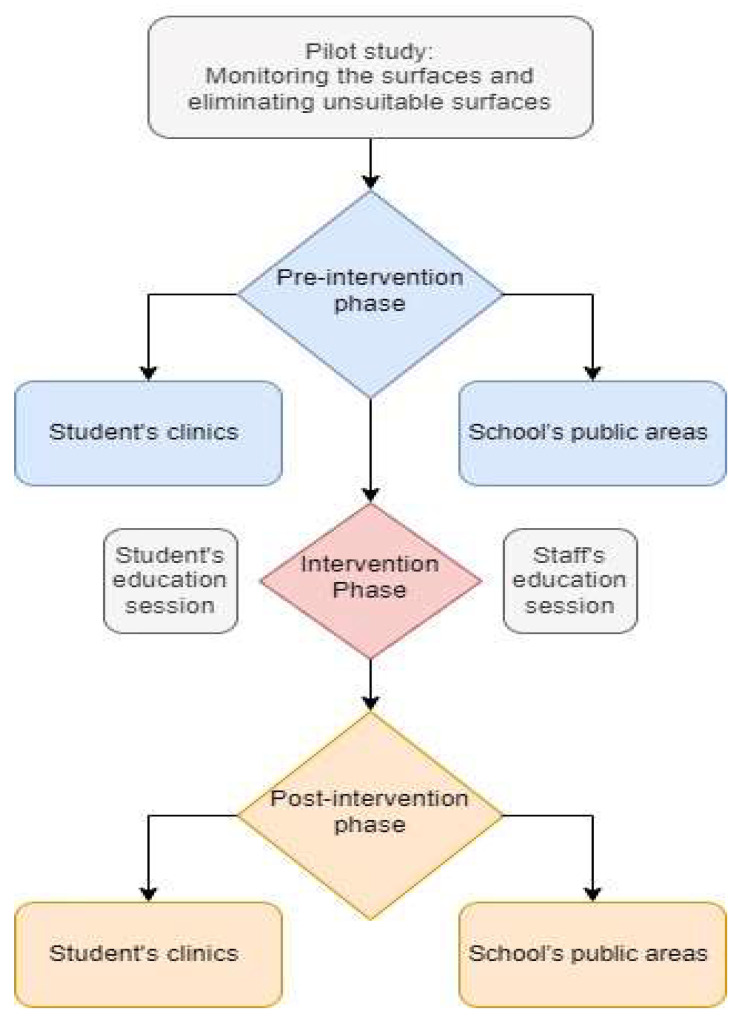
Study design flowchart.

### 2.2. Evaluation of Surface Cleanliness

Fluorescent marker (FM) gel (Glo Germ, Moab, UT, USA) was used. The FM used in the present study is a lotion base containing fluorescent plastic particles (total ingredients are purified water, glyceryl monostearate, ceteareth-20, cetearyl alcohol, white petrolatum, glycerin, organic soybean oil, isopropyl myristate, mineral oil, carbomer 934, disodium EDTA, triethanolamine, methylisothiazolinone, phenoxyethanol, and propylene glycol). The lotion’s use is recommended to avoid cross-contamination and ensure surface cleaning effectiveness, and specifically to avoid the transmission/spread of microbes.

FM was applied with a cotton swab on chosen surfaces in students’ clinics and the school’s public areas (see below). The FM application protocol was based on Dewangan and Gaikwad [31,32]. In brief, FM was applied to an area with a diameter of approximately 1.5 cm to most tested surfaces, except for elongated areas, such as banisters, in which the applied area was approx. 1.5 × 6 cm.

Applications were carried out at the end of a working day, twice a week, for 3 months. Twenty-four hours later, at the end of the following working day, surfaces were examined with a UV flashlight, 51 LEDs, and a 395 nanometer UV light (UV-Rev, Escolite, Dallas, TX, USA). Findings were recorded with a Canon 70D camera (Tokyo, Japan) and photographs were evaluated for the absence or presence of residual gel. The area was considered clean if no FM was detected on the surface (0 FM present). If any FM was visible in the next day’s assessment, the area was marked as unclean/dirty (Figure 2).

### 2.3. Surfaces Examined

In order to choose adequate surfaces to be examined with the use of FM, a pilot study was carried out in which FM was applied to 50 surfaces and examined in the following 24 h. Three surfaces were found to be inadequate for FM use (e.g., LED X-ray apron and wood banisters) and were excluded.

Forty-seven surfaces which were found to be adequate for FM application belonged to the following categories.

#### 2.3.1. Students’ Working Areas

Students’ working cubicles (the students’ cleaning responsibility)—13 points of application (e.g., dental chair handles, countertops, suction handles, unit handles, computer mouse). Before the intervention, 2 random dental units were evaluated each time, at 13 different time points. After the intervention, 3 random dental units were evaluated, at 9 different points of time (Figure 3).

X-ray areas (the students’ cleaning responsibility)—9 points of application (e.g., handle of the examination chair, computer mouse, X-ray conus). The surfaces were evaluated at 13 points of time pre-I and at 9 points of time post-I.

Clinic service areas—3 points of application are as follows: assistants’ desk (the dental assistants’ cleaning responsibility), and external and internal doorknobs (the cleaning responsibility of employees from an external cleaning company). The surfaces were evaluated at 13 points of time pre-I and at 9 points of time post-I.

#### 2.3.2. Public Areas

Public areas in the dental school (the cleaning responsibility of employees of an external cleaning company) (Figure 4). All public surfaces were evaluated at 13 points of time pre-I and at 9 points of time post-I, which are as follows:

Elevators—4 points of application (e.g., internal and external buttons and banisters);

Public areas—10 points of application (e.g., reception desk, chairs, banisters, employees’ time clocks);

Lavatories—9 points of application (e.g., door handles, water flush knobs, tap knobs).

The total number of examined surfaces was 1236. As examinations were carried out during the first year of the COVID-19 pandemic, clinical work was restricted by the Ministry of Health, State of Israel (e.g., working periods, the number of active dental units). Therefore, the number of examined surfaces was different between the two examination periods (574 pre-I and 662 post-I).

Statistical methods: Surface cleanliness for 24 h. After the application, each of the surface categories (students’ dental units, service areas, X-ray areas, elevators, public areas, and lavatories) were calculated as percentages of clean surfaces of the total number of surfaces examined in each category (pre-I and post-I). The higher the percentage, the cleaner the surface.

A one-way ANOVA was performed to evaluate the effect of intervention (the pre-I percentage of clean surfaces compared to the post-I percentage of clean surfaces) for the different surface categories.

## 3. Results

The percentages of clean surfaces in each category are presented in Table 1 and Figure 5.

One-way ANOVA revealed statistically significant differences in surface cleanliness between pre-I and post-I evaluations in the following surface categories: students’ dental cubicles, students’ X-ray areas, and elevators. The difference in the lavatories was borderline.

When mean values of the surfaces in the students’ clinics (students’ cubicles, X-ray areas, and clinic service areas) and the surfaces in the school’s public areas (elevators, public areas, and lavatories) were collapsed and compared to one another, no significant differences between the two groups could be detected in both the pre-I and post-I evaluations. The mean surface cleanliness post-I was 45.08 ± 22.67 for students’ clinics and 47.32 ± 15.04 for the school’s public areas (Figure 6). The difference between pre- and post-intervention values was significant (ANOVA with repeated measures; mean square = 2645.223, F_(1)_ = 10.89, *p* < 0.005).

## 4. Discussion

In the Summary of Infection Prevention Practices in Dental Settings [14], the CDC refers (among others) to environmental infection prevention and control. The document emphasizes the importance of cleaning and disinfecting surfaces that are most likely to become contaminated with pathogens, including clinical contact surfaces (e.g., frequently touched surfaces such as light handles, bracket trays, switches on dental units, and computer equipment) in the patient care area. The CDC points out that ongoing education and training of dental healthcare personnel are critical for ensuring that infection prevention policies and procedures are understood and followed. A successful infection prevention program depends, among other things, on developing standard operating procedures, evaluating practices, and providing feedback to the personnel [18].

The need for infection control and for personnel education and training increased substantially during the COVID-19 pandemic when dental treatment became increasingly hazardous to both dental personnel and patients. The ADA recommended cleaning frequently touched services a minimum of four times a day, depending on the patient load, with a combination of detergent/disinfectant wipes to achieve mechanical cleaning [15].

FMs have been used to assess cleanliness levels in clinical settings due to their easy application and ability to provide a quick assessment of cleanliness [33,34]. FM seems to be a useful analog of contaminated bodily fluids because it spreads easily [25]. In a recent study, Dewangan and Gaikwad showed a strong correlation between a liquid detergent with FM properties and the microbiological gold standard, indicating that it could serve as a simple and cost-effective alternative for assessing cleaning practices on a daily basis [31].

In the present study, FM was used as a feedback-based educational tool to reduce environmental contamination in the dental setting. Results showed that the use of FM as a demonstration of surface cleanliness was effective in increasing surfaces’ cleanliness in both students’ clinics and in the school’s public areas. The percentage of completely clean surfaces (0 FM) increased from around 30% pre-I to around 45% post-I. This indicates that the FM demonstration, in conjunction with increased impact given to environmental infection control, was effective in decreasing surface contamination in the dental school.

The effect of intervention was especially prominent among dental students who showed a significant improvement in the cleanliness of surfaces that were their responsibility (students’ working cubicles, dental units, and X-ray areas). The results were less impressive among para-dental personnel (dental assistants and employees of a cleaning company), with only some of the areas under their responsibility (elevators) showing significantly better results post-I as compared to pre-I. This may be due to COVID-19 restrictions which instructed para-dental personnel (including assistants) to minimize their contact with patients. Employees of the cleaning company were allowed to work only after hours when patients were absent from the building. Thus, dental students were in most direct contact with patients, making them more concerned about personal safety. An additional difference between students and the cleaning personnel may have been in the personal protective equipment used. While students were very strict about changing their gloves between patients, cleaning personnel might have used their disposable gloves for longer periods of time. In such cases, FM could have accumulated on the gloves during the day, a fact which might have created “false positive” results. Although the percentage of completely clean surfaces post-I did not exceed 50% (except in the elevators, approx. 60% completely clean), one has to bear in mind that the dichotomous scoring system (clean/unclean) might have biased the results toward lower scores. To ensure better control over surfaces cleanliness, additional measures can be used (such as plastic covers) for elevator buttons, computer keyboards, etc.

The manual cleaning and disinfection of environmental surfaces in healthcare facilities are essential elements of infection prevention programs, especially during the COVID-19 pandemic [31]. The ability to visualize unclean areas through the use of FM has a potential to reduce the hazard of environmental contamination. The use of fluorescent dye to enhance safety demonstrations has been shown to result in the greater self-reporting of behavioral changes in comparison to other educational techniques [35].

Clearly, pre-dosing surfaces with FM and examining the extent to which it remained on the surface 24 h later represent a crude way of examining the potential of fomite contamination risk. Nevertheless, the ability to demonstrate the uncleanliness of surfaces may have a beneficial effect on the subjects’ future cleaning behavior. Dental students are not too experienced in clinical procedures. Therefore, the implementation of standard universal precautions in dental schools is especially important to control cross-infection [36,37]. Emphasizing environmental infection control and educating dental students to apply appropriate infection control strategies enable a healthy environment in which both dental students and patients are protected [38].

Limitations: Fomite contamination/transmission for SARS-CoV-2 has not been unequivocally proven. No study has yet convincingly shown that directly touching contaminated surfaces leads to self-inoculation and infection in the absence of any other exposure from aerosols. It is also noteworthy that FM cannot be used to detect the presence or absence of specific organisms, so its use in pathogen-specific outbreak evaluation is not practical [24]. FM removal represents merely a physical removal of the applied substance, and, in some cases, surfaces which were not adequately clean may still have been effectively disinfected [30]. Moreover, the spread on unclean areas may have occurred from bags, coats, and other fomites, rather than by hands. Furthermore, some of the beneficial effect of the educational intervention might have been due to an increased awareness to cleaning and disinfection, and not necessarily due to the ability to actually see the contamination. Nevertheless, the significant increase in surface cleanliness following the FM demonstration indicates its potential to promote strategies for environmental infection control in dental clinics during and beyond the COVID-19 pandemic.

## 5. Conclusions

Cross-infection in dental clinics represents a major enduring public concern, but it increased significantly during the global pandemic. Academic training, strict control guidelines, and supervision using efficient low-cost instruments are essential for a safe environment for both dental personnel and patients. FM can act as an efficient educational tool to increase awareness of the control of environmental contamination in large clinics with multiple personnel, such as dental schools.

## Figures and Tables

**Figure 2 ijerph-20-05229-f002:**
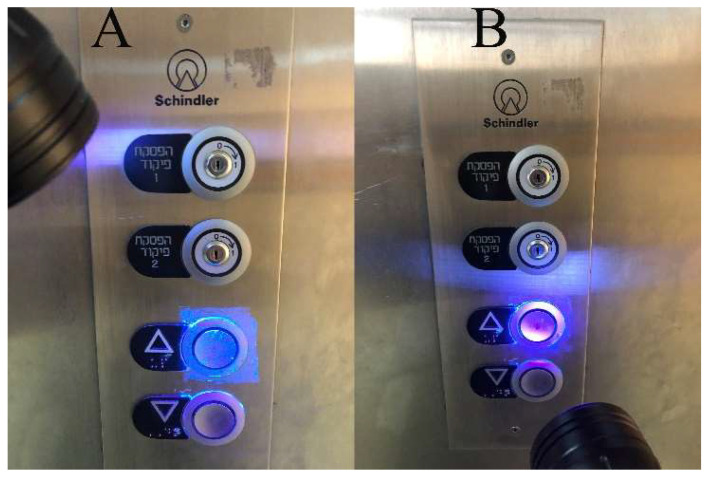
Application of FM to elevator buttons; (**A**): Elevator buttons following FM application; (**B**): elevator buttons 24 h post-application (scored as unclean).

**Figure 3 ijerph-20-05229-f003:**
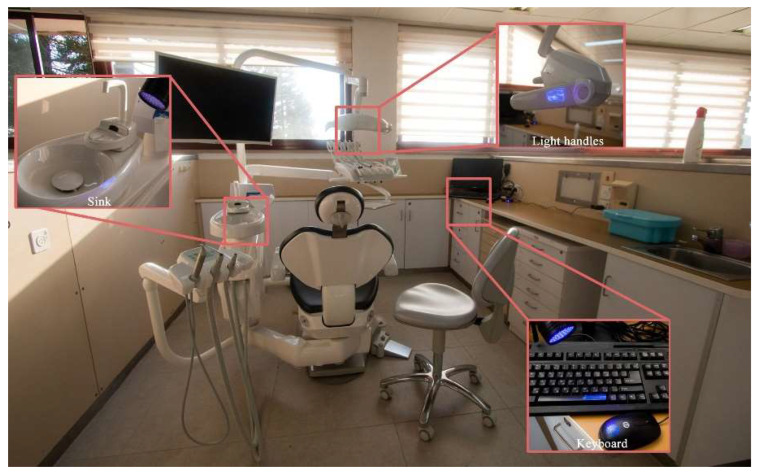
Example of surfaces examined in students’ working cubicles.

**Figure 4 ijerph-20-05229-f004:**
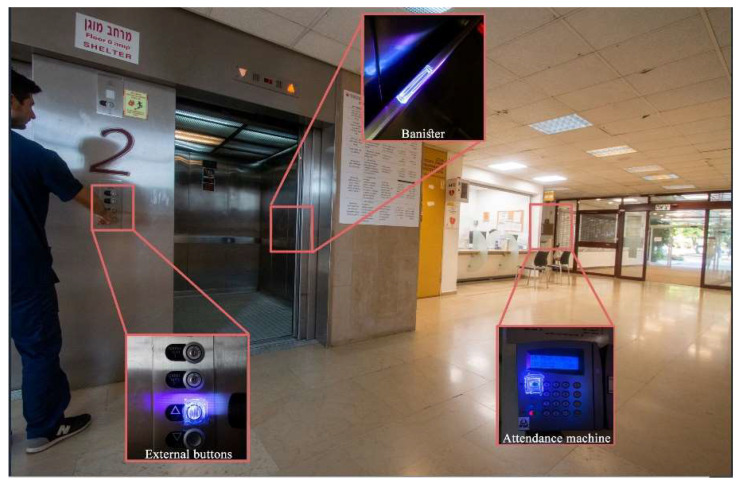
Example of surfaces examined in the school’s public areas.

**Figure 5 ijerph-20-05229-f005:**
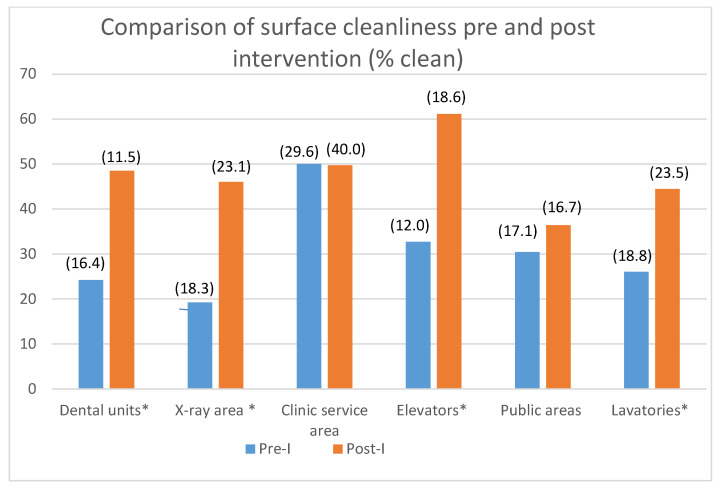
Comparison of surface cleanliness pre- and post-intervention (% clean). * Surfaces in which significant differences were found between pre-I and post-I; numbers in parenthesis represent the standard deviation (SD) values for each bar.

**Figure 6 ijerph-20-05229-f006:**
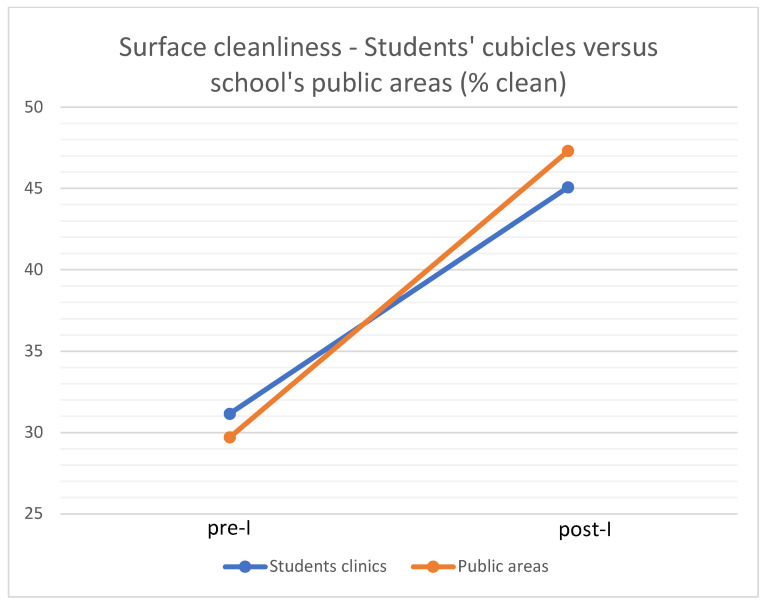
The mean values of surface cleanliness pre-I and post-I in students’ cubicles and in the school’s public areas.

**Table 1 ijerph-20-05229-t001:** The percentage of clean surfaces at pre- and post-intervention evaluations (pre-I versus post-I).

	Cleanliness (%)	Pre-I *	Post-I	ANOVA-F_(1)_	*p* ***
Surface Category					
Students’ cubicles (n = 13) **		24.24 ± 16.14	48.48 ± 11.53	12.41	**0.003**
X-ray areas (n = 9)		19.23 ± 18.33	46.03 ± 23.17	10.03	**0.006**
Clinic service areas (n = 3)		50.00 ± 29.65	49.74 ± 40.06	NS	NS
Elevators (n = 4)		32.69 ± 12.00	61.11 ± 18.16	19.192	**0.000**
Public areas (n = 10)		30.42 ± 17.16	36.41 ± 16.76	NS	NS
Lavatories (n = 9)		26.02 ± 18.87	44.44 ± 23.57	3.717	0.072

* The percentage of clean surfaces (mean ± SD) in each category. ** In parenthesis—the number of examined surfaces in each category. *** Significant differences are marked in bold.

## Data Availability

The data that support the findings of this study are available on request from the corresponding author.

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
