# Peer review of "Fluorescent Marker as a Tool to Improve Strategies to Control Contaminated Surfaces and Decrease Danger of Cross-Contamination in Dental Clinics, during and beyond the COVID-19 Pandemic"

_ijerph, 2023, doi:10.3390/ijerph20065229_

Round 1

Reviewer 1 Report

It is well-established that SARS-COV-2 - more than other respiratory viruses - is spread by aerosols not ‘respiratory droplets’ - though all respiratory viruses are spread by aerosol to some degree:

https://pubmed.ncbi.nlm.nih.gov/35104003/

- so the authors need to review and highlight this in their Introduction to accurately set the scene.

Only after this, should they then discuss the secondary hazard of larger ‘droplets’ settling into surfaces (as humans produce a large range of droplets and aerosols in their everyday speech - see: https://www.nature.com/articles/s41598-019-38808-z).

Also, no study has yet convincingly shown directly that someone touching these contaminated surfaces then goes onto self-inoculate and infect themselves leading to disease - in the absence of any other exposure from aerosols.

So this surface/fomite-contamination/transmission risk has always been theoretical and assumed.

The authors need to recognise and highlight this limitation, which is well-recognised (very reluctantly) in the infection control community.

Finally, it is not very clear exactly what they did - can they create a flowchart/graphic outlining the steps of their study more clearly.

- pre-dosing surfaces with the marker then examining the extent to which it has spread 24 hrs later is a very crude way to examine this potential  fomite-contamination/transmission risk - the spread could also have been done via bags or coats and other fomites - rather than by hands.

The authors need to add this to the study limitations.

Also, if the institution is serious about keeping surfaces clean and easy to disinfect, they can use plastic covers for their elevator buttons, computer keyboards, etc. and clean them regularly - which is what they did in Hong Kong post-SARS-COV-1,  2003. The authors should mention this option also.

Author Response

Dear reviewer,

Thank you for your time and thorough report. We do hope that after addressing all your valuable comments you will find that the manuscript has improved.

COMMENT no. 1: - It is well-established that SARS-COV-2 - more than other respiratory viruses - is spread by aerosols not ‘respiratory droplets’ - though all respiratory viruses are spread by aerosol to some degree:

https://pubmed.ncbi.nlm.nih.gov/35104003/

- so, the authors need to review and highlight this in their Introduction to accurately set the scene.

Only after this, should they then discuss the secondary hazard of larger ‘droplets’ settling into surfaces (as humans produce a large range of droplets and aerosols in their everyday speech - see: https://www.nature.com/articles/s41598-019-38808-z).

Also, no study has yet convincingly shown directly that someone touching these contaminated surfaces then goes onto self-inoculate and infect themselves leading to disease - in the absence of any other exposure from aerosols.

So, this surface/fomite-contamination/transmission risk has always been theoretical and assumed.

The authors need to recognize and highlight this limitation, which is well-recognized (very reluctantly) in the infection control community.”

RESPONSE:  Thank you for this comment, the introduction was expanded to cover all the above-mentioned issues and references were added.

COMMENT no.2: -Finally, it is not very clear exactly what they did - can they create a flowchart/graphic outlining the steps of their study more clearly.

RESPONSE: A flowchart was added as requested to the Material and methods section /study design.

COMMENT no.3: - pre-dosing surfaces with the marker then examining the extent to which it has spread 24 hrs later is a very crude way to examine this potential fomite-contamination/transmission risk - the spread could also have been done via bags or coats and other fomites - rather than by hands.

The authors need to add this to the study limitations.

RESPONSE: A discussion of this issue was added to the text in the limitation section.

COMMENT no.4 : Also, if the institution is serious about keeping surfaces clean and easy to disinfect, they can use plastic covers for their elevator buttons, computer keyboards, etc. and clean them regularly - which is what they did in Hong Kong post-SARS-COV-1, 2003. The authors should mention this option also.

RESPONSE: A suggestion to add plastic covers was added to the discussion part.

Dear reviewer,

Thank you for your time and thorough report. We do hope that after addressing all your valuable comments you will find that the manuscript has improved.

COMMENT no. 1: - It is well-established that SARS-COV-2 - more than other respiratory viruses - is spread by aerosols not ‘respiratory droplets’ - though all respiratory viruses are spread by aerosol to some degree:

https://pubmed.ncbi.nlm.nih.gov/35104003/

- so, the authors need to review and highlight this in their Introduction to accurately set the scene.

Only after this, should they then discuss the secondary hazard of larger ‘droplets’ settling into surfaces (as humans produce a large range of droplets and aerosols in their everyday speech - see: https://www.nature.com/articles/s41598-019-38808-z).

Also, no study has yet convincingly shown directly that someone touching these contaminated surfaces then goes onto self-inoculate and infect themselves leading to disease - in the absence of any other exposure from aerosols.

So, this surface/fomite-contamination/transmission risk has always been theoretical and assumed.

The authors need to recognize and highlight this limitation, which is well-recognized (very reluctantly) in the infection control community.”

RESPONSE:  Thank you for this comment, the introduction was expanded to cover all the above-mentioned issues and references were added.

COMMENT no.2: -Finally, it is not very clear exactly what they did - can they create a flowchart/graphic outlining the steps of their study more clearly.

RESPONSE: A flowchart was added as requested to the Material and methods section /study design.

COMMENT no.3: - pre-dosing surfaces with the marker then examining the extent to which it has spread 24 hrs later is a very crude way to examine this potential fomite-contamination/transmission risk - the spread could also have been done via bags or coats and other fomites - rather than by hands.

The authors need to add this to the study limitations.

RESPONSE: A discussion of this issue was added to the text in the limitation section.

COMMENT no.4 : Also, if the institution is serious about keeping surfaces clean and easy to disinfect, they can use plastic covers for their elevator buttons, computer keyboards, etc. and clean them regularly - which is what they did in Hong Kong post-SARS-COV-1, 2003. The authors should mention this option also.

RESPONSE: A suggestion to add plastic covers was added to the discussion part.

Dear reviewer,

Thank you for your time and thorough report. We do hope that after addressing all your valuable comments you will find that the manuscript has improved.

COMMENT no. 1: - It is well-established that SARS-COV-2 - more than other respiratory viruses - is spread by aerosols not ‘respiratory droplets’ - though all respiratory viruses are spread by aerosol to some degree:

https://pubmed.ncbi.nlm.nih.gov/35104003/

- so, the authors need to review and highlight this in their Introduction to accurately set the scene.

Only after this, should they then discuss the secondary hazard of larger ‘droplets’ settling into surfaces (as humans produce a large range of droplets and aerosols in their everyday speech - see: https://www.nature.com/articles/s41598-019-38808-z).

Also, no study has yet convincingly shown directly that someone touching these contaminated surfaces then goes onto self-inoculate and infect themselves leading to disease - in the absence of any other exposure from aerosols.

So, this surface/fomite-contamination/transmission risk has always been theoretical and assumed.

The authors need to recognize and highlight this limitation, which is well-recognized (very reluctantly) in the infection control community.”

RESPONSE:  Thank you for this comment, the introduction was expanded to cover all the above-mentioned issues and references were added.

COMMENT no.2: -Finally, it is not very clear exactly what they did - can they create a flowchart/graphic outlining the steps of their study more clearly.

RESPONSE: A flowchart was added as requested to the Material and methods section /study design.

COMMENT no.3: - pre-dosing surfaces with the marker then examining the extent to which it has spread 24 hrs later is a very crude way to examine this potential fomite-contamination/transmission risk - the spread could also have been done via bags or coats and other fomites - rather than by hands.

The authors need to add this to the study limitations.

RESPONSE: A discussion of this issue was added to the text in the limitation section.

COMMENT no.4 : Also, if the institution is serious about keeping surfaces clean and easy to disinfect, they can use plastic covers for their elevator buttons, computer keyboards, etc. and clean them regularly - which is what they did in Hong Kong post-SARS-COV-1, 2003. The authors should mention this option also.

RESPONSE: A suggestion to add plastic covers was added to the discussion part.

Dear reviewer,

Thank you for your time and thorough report. We do hope that after addressing all your valuable comments you will find that the manuscript has improved.

COMMENT no. 1: - It is well-established that SARS-COV-2 - more than other respiratory viruses - is spread by aerosols not ‘respiratory droplets’ - though all respiratory viruses are spread by aerosol to some degree:

https://pubmed.ncbi.nlm.nih.gov/35104003/

- so, the authors need to review and highlight this in their Introduction to accurately set the scene.

Only after this, should they then discuss the secondary hazard of larger ‘droplets’ settling into surfaces (as humans produce a large range of droplets and aerosols in their everyday speech - see: https://www.nature.com/articles/s41598-019-38808-z).

Also, no study has yet convincingly shown directly that someone touching these contaminated surfaces then goes onto self-inoculate and infect themselves leading to disease - in the absence of any other exposure from aerosols.

So, this surface/fomite-contamination/transmission risk has always been theoretical and assumed.

The authors need to recognize and highlight this limitation, which is well-recognized (very reluctantly) in the infection control community.”

RESPONSE:  Thank you for this comment, the introduction was expanded to cover all the above-mentioned issues and references were added.

COMMENT no.2: -Finally, it is not very clear exactly what they did - can they create a flowchart/graphic outlining the steps of their study more clearly.

RESPONSE: A flowchart was added as requested to the Material and methods section /study design.

COMMENT no.3: - pre-dosing surfaces with the marker then examining the extent to which it has spread 24 hrs later is a very crude way to examine this potential fomite-contamination/transmission risk - the spread could also have been done via bags or coats and other fomites - rather than by hands.

The authors need to add this to the study limitations.

RESPONSE: A discussion of this issue was added to the text in the limitation section.

COMMENT no.4 : Also, if the institution is serious about keeping surfaces clean and easy to disinfect, they can use plastic covers for their elevator buttons, computer keyboards, etc. and clean them regularly - which is what they did in Hong Kong post-SARS-COV-1, 2003. The authors should mention this option also.

RESPONSE: A suggestion to add plastic covers was added to the discussion part.

Dear reviewer,

Thank you for your time and thorough report. We do hope that after addressing all your valuable comments you will find that the manuscript has improved.

COMMENT no. 1: - It is well-established that SARS-COV-2 - more than other respiratory viruses - is spread by aerosols not ‘respiratory droplets’ - though all respiratory viruses are spread by aerosol to some degree:

https://pubmed.ncbi.nlm.nih.gov/35104003/

- so, the authors need to review and highlight this in their Introduction to accurately set the scene.

Only after this, should they then discuss the secondary hazard of larger ‘droplets’ settling into surfaces (as humans produce a large range of droplets and aerosols in their everyday speech - see: https://www.nature.com/articles/s41598-019-38808-z).

Also, no study has yet convincingly shown directly that someone touching these contaminated surfaces then goes onto self-inoculate and infect themselves leading to disease - in the absence of any other exposure from aerosols.

So, this surface/fomite-contamination/transmission risk has always been theoretical and assumed.

The authors need to recognize and highlight this limitation, which is well-recognized (very reluctantly) in the infection control community.”

RESPONSE:  Thank you for this comment, the introduction was expanded to cover all the above-mentioned issues and references were added.

COMMENT no.2: -Finally, it is not very clear exactly what they did - can they create a flowchart/graphic outlining the steps of their study more clearly.

RESPONSE: A flowchart was added as requested to the Material and methods section /study design.

COMMENT no.3: - pre-dosing surfaces with the marker then examining the extent to which it has spread 24 hrs later is a very crude way to examine this potential fomite-contamination/transmission risk - the spread could also have been done via bags or coats and other fomites - rather than by hands.

The authors need to add this to the study limitations.

RESPONSE: A discussion of this issue was added to the text in the limitation section.

COMMENT no.4 : Also, if the institution is serious about keeping surfaces clean and easy to disinfect, they can use plastic covers for their elevator buttons, computer keyboards, etc. and clean them regularly - which is what they did in Hong Kong post-SARS-COV-1, 2003. The authors should mention this option also.

RESPONSE: A suggestion to add plastic covers was added to the discussion part.

Dear reviewer,

Thank you for your time and thorough report. We do hope that after addressing all your valuable comments you will find that the manuscript has improved.

COMMENT no. 1: - It is well-established that SARS-COV-2 - more than other respiratory viruses - is spread by aerosols not ‘respiratory droplets’ - though all respiratory viruses are spread by aerosol to some degree:

https://pubmed.ncbi.nlm.nih.gov/35104003/

- so, the authors need to review and highlight this in their Introduction to accurately set the scene.

Only after this, should they then discuss the secondary hazard of larger ‘droplets’ settling into surfaces (as humans produce a large range of droplets and aerosols in their everyday speech - see: https://www.nature.com/articles/s41598-019-38808-z).

Also, no study has yet convincingly shown directly that someone touching these contaminated surfaces then goes onto self-inoculate and infect themselves leading to disease - in the absence of any other exposure from aerosols.

So, this surface/fomite-contamination/transmission risk has always been theoretical and assumed.

The authors need to recognize and highlight this limitation, which is well-recognized (very reluctantly) in the infection control community.”

RESPONSE:  Thank you for this comment, the introduction was expanded to cover all the above-mentioned issues and references were added.

COMMENT no.2: -Finally, it is not very clear exactly what they did - can they create a flowchart/graphic outlining the steps of their study more clearly.

RESPONSE: A flowchart was added as requested to the Material and methods section /study design.

COMMENT no.3: - pre-dosing surfaces with the marker then examining the extent to which it has spread 24 hrs later is a very crude way to examine this potential fomite-contamination/transmission risk - the spread could also have been done via bags or coats and other fomites - rather than by hands.

The authors need to add this to the study limitations.

RESPONSE: A discussion of this issue was added to the text in the limitation section.

COMMENT no.4 : Also, if the institution is serious about keeping surfaces clean and easy to disinfect, they can use plastic covers for their elevator buttons, computer keyboards, etc. and clean them regularly - which is what they did in Hong Kong post-SARS-COV-1, 2003. The authors should mention this option also.

RESPONSE: A suggestion to add plastic covers was added to the discussion part.

Reviewer 2 Report

The manuscript entitled “Fluorescent marker as a tool to promote strategies for environ- mental infection control in dental clinics during and beyond  the COVID-19 pandemic” is interesting. However, authors shall revise the following comments before could further consider for publication. Also, the document shall be sent for proofread.

1.     Abstract- Line 21-23. Kindly revise the sentence structure and grammar in this sentence.

2.     Line 30- Citation needed for this claim “the worldwide state of emergency declared in March 2020 and the rapid spread of the COVID-19 pandemic raised an urgent need to develop simple measures to prevent cross-contamination in medical settings, as well as in other public areas.”.

3.     Line 33- Lump references shall be avoided. Maximum of 3 references are recommended for each statement.

4.     Line 40- “intraoral rotating instruments which disperse microscopic airborne particles posed an increased threat to both dental personnel and patients”. As far as the reviewer’s concerned, saliva is mainly related to direct surface contamination instead of airborne (due to the density and size of the droplets). Unless it is inhale/ exhale process. Please provide some literature to support your claim.

5.     Line 58-60- The information on surface cleanliness in healthcare-related facilities is very limited. Only one example (fluorescent gel) was mentioned. Sodium hypochlorite solution, an EPA-approved surface disinfectant for healthcare facilities shall be included to improve the content of the present manuscript. Such information is widely reported. Example: https://doi.org/10.1007/s11356-021-16171-9

6.     Line 74-“ The aim of the present study was twofold”. Please revised this sentence. Fragmented sentence.

7.     Authors shall highlight the discrepancy of past works, and then highlight the gap that this manuscript has addressed.

8.     Pictures 1-3 shall be labelled. For example in Fig. 3, sink, keyboard, etc. Also, why start with picture instead of figure? Usually, a manuscript only contains of figures, tables, or supplementary documents. Less likely to use pictures as the caption. PLease check the author's guideline of IJERPH

9.     Line 149- Past study used similar samplings no. for pre- and post-examination. Authors shall justify why this study considered a different number of pre-and post-examinations.

10. Plot of SD on each bar shall be included in Figure 1

11. Correlation of surface cleanliness- students cubicles vs. school’s public areas shall be included in Figure 2.

12. Discussion- Line 179- which “Guideline” that authors referred to? Kindly name it and includes it as a reference.

13. The distribution of paragraphs is not consistent. (Line 218-221). Does this paragraph only consist of 1 sentence? Please merge with other paragraphs.

14. Kindly include recent 5 years' references to show that your manuscript is up-to-date.

Author Response

Dear reviewer,

Thank you for your time and thorough report. We do hope that after addressing all your valuable comments you will find that the manuscript has improved.

The manuscript entitled “Fluorescent marker as a tool to promote strategies for environ- mental infection control in dental clinics during and beyond the COVID-19 pandemic” is interesting. However, authors shall revise the following comments before could further consider for publication. Also, the document shall be sent for proofread.

  1. Abstract- Line 21-23. Kindly revise the sentence structure and grammar in this sentence.

RESPONSE: Sentence was revised.

  1. Line 30- Citation needed for this claim “the worldwide state of emergency declared in March 2020 and the rapid spread of the COVID-19 pandemic raised an urgent need to develop simple measures to prevent cross-contamination in medical settings, as well as in other public areas.”.

RESPONSE: Citation was added as requested.

  1. Line 33- Lump references shall be avoided. A maximum of 3 references are recommended for each statement.

RESPONSE: As suggested, some of the references were omitted.

  1. Line 40- “intraoral rotating instruments which disperse microscopic airborne particles posed an increased threat to both dental personnel and patients”. As far as the reviewer’s concerned, saliva is mainly related to direct surface contamination instead of airborne (due to the density and size of the droplets). Unless it is inhale/ exhale process.Please provide some literature to support your claim.

RESPONSE: Thank you for this comment. The Introduction section was substantially expanded to cover this issue.

  1. Line 58-60- The information on surface cleanliness in healthcare-related facilities is very limited. Only one example (fluorescent gel) was mentioned. Sodium hypochlorite solution, an EPA-approved surface disinfectant for healthcare facilities shall be included to improve the content of the present manuscript. Such information is widely reported. Example: https://doi.org/10.1007/s11356-021-16171-9

RESPONSE: Information as well as the reference were added to the text.

  1. Line 74-“The aim of the present study was twofold”. Please revise this sentence. Fragmented sentence.

RESPONSE: The sentence was revised.

  1. Authors shall highlight the discrepancy of past works, and then highlight the gap that this manuscript has addressed.

RESPONSE: New references were added.

  1. Pictures 1-3 shall be labelled. For example in Fig. 3, sink, keyboard, etc. Also, why start with picture instead of figure? Usually, a manuscript only contains of figures, tables, or supplementary documents. Less likely to use pictures as the caption. PLease check the author's guideline of IJERPH

RESPONSE: Labels were added to the figures.

  1. Line 149- Past study used similar samplings no. for pre- and post-examination. Authors shall justify why this study considered a different number of pre-and post-examinations.

RESPONSE: Explanation was added to the text:

As examinations were carried out during the first year of the COVID-19 pandemic, clinical work was restricted by the Ministry of Health, State of Israel (e.g., working periods, the number of active dental units, etc.). Therefore, the number of examined surfaces was different between the two examination periods (574 pre-I and 662 post-I).

  1. Plot of SD on each bar shall be included in Figure 1

RESPONSE: Numbers indicating SD values were added at the top of the bars.

  1. Correlation of surface cleanliness- students cubicles vs. school’s public areas shall be included in Figure 2.

RESPONSE: The figure presents results of a ANOVA with repeated measures: Within-Subjects Factors Public areas vs. Students’ cubicle; between subjects factor Pre I vs. Post I. As the only main effect was between Pre I and Post I findings only these results are quoted in the text (ANOVA, Mean square = 2645.223, F (1) = 10.89, p<0.005).

  1. Discussion- Line 179- which “Guideline” that authors referred to? Kindly name it and includes it as a reference.

RESPONSE: Reference was added.

  1. The distribution of paragraphs is not consistent. (Line 218-221). Does this paragraph only consist of 1 sentence? Please merge with other paragraphs.

RESPONSE: Paragraphs were re-organized as recommended.

  1. Kindly include the last 5 years' references to show that your manuscript is up-to-date.

RESPONSE: Recent references were added.

The manuscript underwent English editing by MPDI English editing service. Certificate was sent to the Editor.

Dear reviewer,

Thank you for your time and thorough report. We do hope that after addressing all your valuable comments you will find that the manuscript has improved.

The manuscript entitled “Fluorescent marker as a tool to promote strategies for environ- mental infection control in dental clinics during and beyond the COVID-19 pandemic” is interesting. However, authors shall revise the following comments before could further consider for publication. Also, the document shall be sent for proofread.

  1. Abstract- Line 21-23. Kindly revise the sentence structure and grammar in this sentence.

RESPONSE: Sentence was revised.

  1. Line 30- Citation needed for this claim “the worldwide state of emergency declared in March 2020 and the rapid spread of the COVID-19 pandemic raised an urgent need to develop simple measures to prevent cross-contamination in medical settings, as well as in other public areas.”.

RESPONSE: Citation was added as requested.

  1. Line 33- Lump references shall be avoided. A maximum of 3 references are recommended for each statement.

RESPONSE: As suggested, some of the references were omitted.

  1. Line 40- “intraoral rotating instruments which disperse microscopic airborne particles posed an increased threat to both dental personnel and patients”. As far as the reviewer’s concerned, saliva is mainly related to direct surface contamination instead of airborne (due to the density and size of the droplets). Unless it is inhale/ exhale process.Please provide some literature to support your claim.

RESPONSE: Thank you for this comment. The Introduction section was substantially expanded to cover this issue.

  1. Line 58-60- The information on surface cleanliness in healthcare-related facilities is very limited. Only one example (fluorescent gel) was mentioned. Sodium hypochlorite solution, an EPA-approved surface disinfectant for healthcare facilities shall be included to improve the content of the present manuscript. Such information is widely reported. Example: https://doi.org/10.1007/s11356-021-16171-9

RESPONSE: Information as well as the reference were added to the text.

  1. Line 74-“The aim of the present study was twofold”. Please revise this sentence. Fragmented sentence.

RESPONSE: The sentence was revised.

  1. Authors shall highlight the discrepancy of past works, and then highlight the gap that this manuscript has addressed.

RESPONSE: New references were added.

  1. Pictures 1-3 shall be labelled. For example in Fig. 3, sink, keyboard, etc. Also, why start with picture instead of figure? Usually, a manuscript only contains of figures, tables, or supplementary documents. Less likely to use pictures as the caption. PLease check the author's guideline of IJERPH

RESPONSE: Labels were added to the figures.

  1. Line 149- Past study used similar samplings no. for pre- and post-examination. Authors shall justify why this study considered a different number of pre-and post-examinations.

RESPONSE: Explanation was added to the text:

As examinations were carried out during the first year of the COVID-19 pandemic, clinical work was restricted by the Ministry of Health, State of Israel (e.g., working periods, the number of active dental units, etc.). Therefore, the number of examined surfaces was different between the two examination periods (574 pre-I and 662 post-I).

  1. Plot of SD on each bar shall be included in Figure 1

RESPONSE: Numbers indicating SD values were added at the top of the bars.

  1. Correlation of surface cleanliness- students cubicles vs. school’s public areas shall be included in Figure 2.

RESPONSE: The figure presents results of a ANOVA with repeated measures: Within-Subjects Factors Public areas vs. Students’ cubicle; between subjects factor Pre I vs. Post I. As the only main effect was between Pre I and Post I findings only these results are quoted in the text (ANOVA, Mean square = 2645.223, F (1) = 10.89, p<0.005).

  1. Discussion- Line 179- which “Guideline” that authors referred to? Kindly name it and includes it as a reference.

RESPONSE: Reference was added.

  1. The distribution of paragraphs is not consistent. (Line 218-221). Does this paragraph only consist of 1 sentence? Please merge with other paragraphs.

RESPONSE: Paragraphs were re-organized as recommended.

  1. Kindly include the last 5 years' references to show that your manuscript is up-to-date.

RESPONSE: Recent references were added.

The manuscript underwent English editing by MPDI English editing service. Certificate was sent to the Editor.

Dear reviewer,

Thank you for your time and thorough report. We do hope that after addressing all your valuable comments you will find that the manuscript has improved.

The manuscript entitled “Fluorescent marker as a tool to promote strategies for environ- mental infection control in dental clinics during and beyond the COVID-19 pandemic” is interesting. However, authors shall revise the following comments before could further consider for publication. Also, the document shall be sent for proofread.

  1. Abstract- Line 21-23. Kindly revise the sentence structure and grammar in this sentence.

RESPONSE: Sentence was revised.

  1. Line 30- Citation needed for this claim “the worldwide state of emergency declared in March 2020 and the rapid spread of the COVID-19 pandemic raised an urgent need to develop simple measures to prevent cross-contamination in medical settings, as well as in other public areas.”.

RESPONSE: Citation was added as requested.

  1. Line 33- Lump references shall be avoided. A maximum of 3 references are recommended for each statement.

RESPONSE: As suggested, some of the references were omitted.

  1. Line 40- “intraoral rotating instruments which disperse microscopic airborne particles posed an increased threat to both dental personnel and patients”. As far as the reviewer’s concerned, saliva is mainly related to direct surface contamination instead of airborne (due to the density and size of the droplets). Unless it is inhale/ exhale process.Please provide some literature to support your claim.

RESPONSE: Thank you for this comment. The Introduction section was substantially expanded to cover this issue.

  1. Line 58-60- The information on surface cleanliness in healthcare-related facilities is very limited. Only one example (fluorescent gel) was mentioned. Sodium hypochlorite solution, an EPA-approved surface disinfectant for healthcare facilities shall be included to improve the content of the present manuscript. Such information is widely reported. Example: https://doi.org/10.1007/s11356-021-16171-9

RESPONSE: Information as well as the reference were added to the text.

  1. Line 74-“The aim of the present study was twofold”. Please revise this sentence. Fragmented sentence.

RESPONSE: The sentence was revised.

  1. Authors shall highlight the discrepancy of past works, and then highlight the gap that this manuscript has addressed.

RESPONSE: New references were added.

  1. Pictures 1-3 shall be labelled. For example in Fig. 3, sink, keyboard, etc. Also, why start with picture instead of figure? Usually, a manuscript only contains of figures, tables, or supplementary documents. Less likely to use pictures as the caption. PLease check the author's guideline of IJERPH

RESPONSE: Labels were added to the figures.

  1. Line 149- Past study used similar samplings no. for pre- and post-examination. Authors shall justify why this study considered a different number of pre-and post-examinations.

RESPONSE: Explanation was added to the text:

As examinations were carried out during the first year of the COVID-19 pandemic, clinical work was restricted by the Ministry of Health, State of Israel (e.g., working periods, the number of active dental units, etc.). Therefore, the number of examined surfaces was different between the two examination periods (574 pre-I and 662 post-I).

  1. Plot of SD on each bar shall be included in Figure 1

RESPONSE: Numbers indicating SD values were added at the top of the bars.

  1. Correlation of surface cleanliness- students cubicles vs. school’s public areas shall be included in Figure 2.

RESPONSE: The figure presents results of a ANOVA with repeated measures: Within-Subjects Factors Public areas vs. Students’ cubicle; between subjects factor Pre I vs. Post I. As the only main effect was between Pre I and Post I findings only these results are quoted in the text (ANOVA, Mean square = 2645.223, F (1) = 10.89, p<0.005).

  1. Discussion- Line 179- which “Guideline” that authors referred to? Kindly name it and includes it as a reference.

RESPONSE: Reference was added.

  1. The distribution of paragraphs is not consistent. (Line 218-221). Does this paragraph only consist of 1 sentence? Please merge with other paragraphs.

RESPONSE: Paragraphs were re-organized as recommended.

  1. Kindly include the last 5 years' references to show that your manuscript is up-to-date.

RESPONSE: Recent references were added.

The manuscript underwent English editing by MPDI English editing service. Certificate was sent to the Editor.

Dear reviewer,

Thank you for your time and thorough report. We do hope that after addressing all your valuable comments you will find that the manuscript has improved.

The manuscript entitled “Fluorescent marker as a tool to promote strategies for environ- mental infection control in dental clinics during and beyond the COVID-19 pandemic” is interesting. However, authors shall revise the following comments before could further consider for publication. Also, the document shall be sent for proofread.

  1. Abstract- Line 21-23. Kindly revise the sentence structure and grammar in this sentence.

RESPONSE: Sentence was revised.

  1. Line 30- Citation needed for this claim “the worldwide state of emergency declared in March 2020 and the rapid spread of the COVID-19 pandemic raised an urgent need to develop simple measures to prevent cross-contamination in medical settings, as well as in other public areas.”.

RESPONSE: Citation was added as requested.

  1. Line 33- Lump references shall be avoided. A maximum of 3 references are recommended for each statement.

RESPONSE: As suggested, some of the references were omitted.

  1. Line 40- “intraoral rotating instruments which disperse microscopic airborne particles posed an increased threat to both dental personnel and patients”. As far as the reviewer’s concerned, saliva is mainly related to direct surface contamination instead of airborne (due to the density and size of the droplets). Unless it is inhale/ exhale process.Please provide some literature to support your claim.

RESPONSE: Thank you for this comment. The Introduction section was substantially expanded to cover this issue.

  1. Line 58-60- The information on surface cleanliness in healthcare-related facilities is very limited. Only one example (fluorescent gel) was mentioned. Sodium hypochlorite solution, an EPA-approved surface disinfectant for healthcare facilities shall be included to improve the content of the present manuscript. Such information is widely reported. Example: https://doi.org/10.1007/s11356-021-16171-9

RESPONSE: Information as well as the reference were added to the text.

  1. Line 74-“The aim of the present study was twofold”. Please revise this sentence. Fragmented sentence.

RESPONSE: The sentence was revised.

  1. Authors shall highlight the discrepancy of past works, and then highlight the gap that this manuscript has addressed.

RESPONSE: New references were added.

  1. Pictures 1-3 shall be labelled. For example in Fig. 3, sink, keyboard, etc. Also, why start with picture instead of figure? Usually, a manuscript only contains of figures, tables, or supplementary documents. Less likely to use pictures as the caption. PLease check the author's guideline of IJERPH

RESPONSE: Labels were added to the figures.

  1. Line 149- Past study used similar samplings no. for pre- and post-examination. Authors shall justify why this study considered a different number of pre-and post-examinations.

RESPONSE: Explanation was added to the text:

As examinations were carried out during the first year of the COVID-19 pandemic, clinical work was restricted by the Ministry of Health, State of Israel (e.g., working periods, the number of active dental units, etc.). Therefore, the number of examined surfaces was different between the two examination periods (574 pre-I and 662 post-I).

  1. Plot of SD on each bar shall be included in Figure 1

RESPONSE: Numbers indicating SD values were added at the top of the bars.

  1. Correlation of surface cleanliness- students cubicles vs. school’s public areas shall be included in Figure 2.

RESPONSE: The figure presents results of a ANOVA with repeated measures: Within-Subjects Factors Public areas vs. Students’ cubicle; between subjects factor Pre I vs. Post I. As the only main effect was between Pre I and Post I findings only these results are quoted in the text (ANOVA, Mean square = 2645.223, F (1) = 10.89, p<0.005).

  1. Discussion- Line 179- which “Guideline” that authors referred to? Kindly name it and includes it as a reference.

RESPONSE: Reference was added.

  1. The distribution of paragraphs is not consistent. (Line 218-221). Does this paragraph only consist of 1 sentence? Please merge with other paragraphs.

RESPONSE: Paragraphs were re-organized as recommended.

  1. Kindly include the last 5 years' references to show that your manuscript is up-to-date.

RESPONSE: Recent references were added.

The manuscript underwent English editing by MPDI English editing service. Certificate was sent to the Editor.

Dear reviewer,

Thank you for your time and thorough report. We do hope that after addressing all your valuable comments you will find that the manuscript has improved.

The manuscript entitled “Fluorescent marker as a tool to promote strategies for environ- mental infection control in dental clinics during and beyond the COVID-19 pandemic” is interesting. However, authors shall revise the following comments before could further consider for publication. Also, the document shall be sent for proofread.

  1. Abstract- Line 21-23. Kindly revise the sentence structure and grammar in this sentence.

RESPONSE: Sentence was revised.

  1. Line 30- Citation needed for this claim “the worldwide state of emergency declared in March 2020 and the rapid spread of the COVID-19 pandemic raised an urgent need to develop simple measures to prevent cross-contamination in medical settings, as well as in other public areas.”.

RESPONSE: Citation was added as requested.

  1. Line 33- Lump references shall be avoided. A maximum of 3 references are recommended for each statement.

RESPONSE: As suggested, some of the references were omitted.

  1. Line 40- “intraoral rotating instruments which disperse microscopic airborne particles posed an increased threat to both dental personnel and patients”. As far as the reviewer’s concerned, saliva is mainly related to direct surface contamination instead of airborne (due to the density and size of the droplets). Unless it is inhale/ exhale process.Please provide some literature to support your claim.

RESPONSE: Thank you for this comment. The Introduction section was substantially expanded to cover this issue.

  1. Line 58-60- The information on surface cleanliness in healthcare-related facilities is very limited. Only one example (fluorescent gel) was mentioned. Sodium hypochlorite solution, an EPA-approved surface disinfectant for healthcare facilities shall be included to improve the content of the present manuscript. Such information is widely reported. Example: https://doi.org/10.1007/s11356-021-16171-9

RESPONSE: Information as well as the reference were added to the text.

  1. Line 74-“The aim of the present study was twofold”. Please revise this sentence. Fragmented sentence.

RESPONSE: The sentence was revised.

  1. Authors shall highlight the discrepancy of past works, and then highlight the gap that this manuscript has addressed.

RESPONSE: New references were added.

  1. Pictures 1-3 shall be labelled. For example in Fig. 3, sink, keyboard, etc. Also, why start with picture instead of figure? Usually, a manuscript only contains of figures, tables, or supplementary documents. Less likely to use pictures as the caption. PLease check the author's guideline of IJERPH

RESPONSE: Labels were added to the figures.

  1. Line 149- Past study used similar samplings no. for pre- and post-examination. Authors shall justify why this study considered a different number of pre-and post-examinations.

RESPONSE: Explanation was added to the text:

As examinations were carried out during the first year of the COVID-19 pandemic, clinical work was restricted by the Ministry of Health, State of Israel (e.g., working periods, the number of active dental units, etc.). Therefore, the number of examined surfaces was different between the two examination periods (574 pre-I and 662 post-I).

  1. Plot of SD on each bar shall be included in Figure 1

RESPONSE: Numbers indicating SD values were added at the top of the bars.

  1. Correlation of surface cleanliness- students cubicles vs. school’s public areas shall be included in Figure 2.

RESPONSE: The figure presents results of a ANOVA with repeated measures: Within-Subjects Factors Public areas vs. Students’ cubicle; between subjects factor Pre I vs. Post I. As the only main effect was between Pre I and Post I findings only these results are quoted in the text (ANOVA, Mean square = 2645.223, F (1) = 10.89, p<0.005).

  1. Discussion- Line 179- which “Guideline” that authors referred to? Kindly name it and includes it as a reference.

RESPONSE: Reference was added.

  1. The distribution of paragraphs is not consistent. (Line 218-221). Does this paragraph only consist of 1 sentence? Please merge with other paragraphs.

RESPONSE: Paragraphs were re-organized as recommended.

  1. Kindly include the last 5 years' references to show that your manuscript is up-to-date.

RESPONSE: Recent references were added.

The manuscript underwent English editing by MPDI English editing service. Certificate was sent to the Editor.

Dear reviewer,

Thank you for your time and thorough report. We do hope that after addressing all your valuable comments you will find that the manuscript has improved.

The manuscript entitled “Fluorescent marker as a tool to promote strategies for environ- mental infection control in dental clinics during and beyond the COVID-19 pandemic” is interesting. However, authors shall revise the following comments before could further consider for publication. Also, the document shall be sent for proofread.

  1. Abstract- Line 21-23. Kindly revise the sentence structure and grammar in this sentence.

RESPONSE: Sentence was revised.

  1. Line 30- Citation needed for this claim “the worldwide state of emergency declared in March 2020 and the rapid spread of the COVID-19 pandemic raised an urgent need to develop simple measures to prevent cross-contamination in medical settings, as well as in other public areas.”.

RESPONSE: Citation was added as requested.

  1. Line 33- Lump references shall be avoided. A maximum of 3 references are recommended for each statement.

RESPONSE: As suggested, some of the references were omitted.

  1. Line 40- “intraoral rotating instruments which disperse microscopic airborne particles posed an increased threat to both dental personnel and patients”. As far as the reviewer’s concerned, saliva is mainly related to direct surface contamination instead of airborne (due to the density and size of the droplets). Unless it is inhale/ exhale process.Please provide some literature to support your claim.

RESPONSE: Thank you for this comment. The Introduction section was substantially expanded to cover this issue.

  1. Line 58-60- The information on surface cleanliness in healthcare-related facilities is very limited. Only one example (fluorescent gel) was mentioned. Sodium hypochlorite solution, an EPA-approved surface disinfectant for healthcare facilities shall be included to improve the content of the present manuscript. Such information is widely reported. Example: https://doi.org/10.1007/s11356-021-16171-9

RESPONSE: Information as well as the reference were added to the text.

  1. Line 74-“The aim of the present study was twofold”. Please revise this sentence. Fragmented sentence.

RESPONSE: The sentence was revised.

  1. Authors shall highlight the discrepancy of past works, and then highlight the gap that this manuscript has addressed.

RESPONSE: New references were added.

  1. Pictures 1-3 shall be labelled. For example in Fig. 3, sink, keyboard, etc. Also, why start with picture instead of figure? Usually, a manuscript only contains of figures, tables, or supplementary documents. Less likely to use pictures as the caption. PLease check the author's guideline of IJERPH

RESPONSE: Labels were added to the figures.

  1. Line 149- Past study used similar samplings no. for pre- and post-examination. Authors shall justify why this study considered a different number of pre-and post-examinations.

RESPONSE: Explanation was added to the text:

As examinations were carried out during the first year of the COVID-19 pandemic, clinical work was restricted by the Ministry of Health, State of Israel (e.g., working periods, the number of active dental units, etc.). Therefore, the number of examined surfaces was different between the two examination periods (574 pre-I and 662 post-I).

  1. Plot of SD on each bar shall be included in Figure 1

RESPONSE: Numbers indicating SD values were added at the top of the bars.

  1. Correlation of surface cleanliness- students cubicles vs. school’s public areas shall be included in Figure 2.

RESPONSE: The figure presents results of a ANOVA with repeated measures: Within-Subjects Factors Public areas vs. Students’ cubicle; between subjects factor Pre I vs. Post I. As the only main effect was between Pre I and Post I findings only these results are quoted in the text (ANOVA, Mean square = 2645.223, F (1) = 10.89, p<0.005).

  1. Discussion- Line 179- which “Guideline” that authors referred to? Kindly name it and includes it as a reference.

RESPONSE: Reference was added.

  1. The distribution of paragraphs is not consistent. (Line 218-221). Does this paragraph only consist of 1 sentence? Please merge with other paragraphs.

RESPONSE: Paragraphs were re-organized as recommended.

  1. Kindly include the last 5 years' references to show that your manuscript is up-to-date.

RESPONSE: Recent references were added.

The manuscript underwent English editing by MPDI English editing service. Certificate was sent to the Editor.

Dear reviewer,

Thank you for your time and thorough report. We do hope that after addressing all your valuable comments you will find that the manuscript has improved.

The manuscript entitled “Fluorescent marker as a tool to promote strategies for environ- mental infection control in dental clinics during and beyond the COVID-19 pandemic” is interesting. However, authors shall revise the following comments before could further consider for publication. Also, the document shall be sent for proofread.

  1. Abstract- Line 21-23. Kindly revise the sentence structure and grammar in this sentence.

RESPONSE: Sentence was revised.

  1. Line 30- Citation needed for this claim “the worldwide state of emergency declared in March 2020 and the rapid spread of the COVID-19 pandemic raised an urgent need to develop simple measures to prevent cross-contamination in medical settings, as well as in other public areas.”.

RESPONSE: Citation was added as requested.

  1. Line 33- Lump references shall be avoided. A maximum of 3 references are recommended for each statement.

RESPONSE: As suggested, some of the references were omitted.

  1. Line 40- “intraoral rotating instruments which disperse microscopic airborne particles posed an increased threat to both dental personnel and patients”. As far as the reviewer’s concerned, saliva is mainly related to direct surface contamination instead of airborne (due to the density and size of the droplets). Unless it is inhale/ exhale process.Please provide some literature to support your claim.

RESPONSE: Thank you for this comment. The Introduction section was substantially expanded to cover this issue.

  1. Line 58-60- The information on surface cleanliness in healthcare-related facilities is very limited. Only one example (fluorescent gel) was mentioned. Sodium hypochlorite solution, an EPA-approved surface disinfectant for healthcare facilities shall be included to improve the content of the present manuscript. Such information is widely reported. Example: https://doi.org/10.1007/s11356-021-16171-9

RESPONSE: Information as well as the reference were added to the text.

  1. Line 74-“The aim of the present study was twofold”. Please revise this sentence. Fragmented sentence.

RESPONSE: The sentence was revised.

  1. Authors shall highlight the discrepancy of past works, and then highlight the gap that this manuscript has addressed.

RESPONSE: New references were added.

  1. Pictures 1-3 shall be labelled. For example in Fig. 3, sink, keyboard, etc. Also, why start with picture instead of figure? Usually, a manuscript only contains of figures, tables, or supplementary documents. Less likely to use pictures as the caption. PLease check the author's guideline of IJERPH

RESPONSE: Labels were added to the figures.

  1. Line 149- Past study used similar samplings no. for pre- and post-examination. Authors shall justify why this study considered a different number of pre-and post-examinations.

RESPONSE: Explanation was added to the text:

As examinations were carried out during the first year of the COVID-19 pandemic, clinical work was restricted by the Ministry of Health, State of Israel (e.g., working periods, the number of active dental units, etc.). Therefore, the number of examined surfaces was different between the two examination periods (574 pre-I and 662 post-I).

  1. Plot of SD on each bar shall be included in Figure 1

RESPONSE: Numbers indicating SD values were added at the top of the bars.

  1. Correlation of surface cleanliness- students cubicles vs. school’s public areas shall be included in Figure 2.

RESPONSE: The figure presents results of a ANOVA with repeated measures: Within-Subjects Factors Public areas vs. Students’ cubicle; between subjects factor Pre I vs. Post I. As the only main effect was between Pre I and Post I findings only these results are quoted in the text (ANOVA, Mean square = 2645.223, F (1) = 10.89, p<0.005).

  1. Discussion- Line 179- which “Guideline” that authors referred to? Kindly name it and includes it as a reference.

RESPONSE: Reference was added.

  1. The distribution of paragraphs is not consistent. (Line 218-221). Does this paragraph only consist of 1 sentence? Please merge with other paragraphs.

RESPONSE: Paragraphs were re-organized as recommended.

  1. Kindly include the last 5 years' references to show that your manuscript is up-to-date.

RESPONSE: Recent references were added.

The manuscript underwent English editing by MPDI English editing service. Certificate was sent to the Editor.

Dear reviewer,

Thank you for your time and thorough report. We do hope that after addressing all your valuable comments you will find that the manuscript has improved.

The manuscript entitled “Fluorescent marker as a tool to promote strategies for environ- mental infection control in dental clinics during and beyond the COVID-19 pandemic” is interesting. However, authors shall revise the following comments before could further consider for publication. Also, the document shall be sent for proofread.

  1. Abstract- Line 21-23. Kindly revise the sentence structure and grammar in this sentence.

RESPONSE: Sentence was revised.

  1. Line 30- Citation needed for this claim “the worldwide state of emergency declared in March 2020 and the rapid spread of the COVID-19 pandemic raised an urgent need to develop simple measures to prevent cross-contamination in medical settings, as well as in other public areas.”.

RESPONSE: Citation was added as requested.

  1. Line 33- Lump references shall be avoided. A maximum of 3 references are recommended for each statement.

RESPONSE: As suggested, some of the references were omitted.

  1. Line 40- “intraoral rotating instruments which disperse microscopic airborne particles posed an increased threat to both dental personnel and patients”. As far as the reviewer’s concerned, saliva is mainly related to direct surface contamination instead of airborne (due to the density and size of the droplets). Unless it is inhale/ exhale process.Please provide some literature to support your claim.

RESPONSE: Thank you for this comment. The Introduction section was substantially expanded to cover this issue.

  1. Line 58-60- The information on surface cleanliness in healthcare-related facilities is very limited. Only one example (fluorescent gel) was mentioned. Sodium hypochlorite solution, an EPA-approved surface disinfectant for healthcare facilities shall be included to improve the content of the present manuscript. Such information is widely reported. Example: https://doi.org/10.1007/s11356-021-16171-9

RESPONSE: Information as well as the reference were added to the text.

  1. Line 74-“The aim of the present study was twofold”. Please revise this sentence. Fragmented sentence.

RESPONSE: The sentence was revised.

  1. Authors shall highlight the discrepancy of past works, and then highlight the gap that this manuscript has addressed.

RESPONSE: New references were added.

  1. Pictures 1-3 shall be labelled. For example in Fig. 3, sink, keyboard, etc. Also, why start with picture instead of figure? Usually, a manuscript only contains of figures, tables, or supplementary documents. Less likely to use pictures as the caption. PLease check the author's guideline of IJERPH

RESPONSE: Labels were added to the figures.

  1. Line 149- Past study used similar samplings no. for pre- and post-examination. Authors shall justify why this study considered a different number of pre-and post-examinations.

RESPONSE: Explanation was added to the text:

As examinations were carried out during the first year of the COVID-19 pandemic, clinical work was restricted by the Ministry of Health, State of Israel (e.g., working periods, the number of active dental units, etc.). Therefore, the number of examined surfaces was different between the two examination periods (574 pre-I and 662 post-I).

  1. Plot of SD on each bar shall be included in Figure 1

RESPONSE: Numbers indicating SD values were added at the top of the bars.

  1. Correlation of surface cleanliness- students cubicles vs. school’s public areas shall be included in Figure 2.

RESPONSE: The figure presents results of a ANOVA with repeated measures: Within-Subjects Factors Public areas vs. Students’ cubicle; between subjects factor Pre I vs. Post I. As the only main effect was between Pre I and Post I findings only these results are quoted in the text (ANOVA, Mean square = 2645.223, F (1) = 10.89, p<0.005).

  1. Discussion- Line 179- which “Guideline” that authors referred to? Kindly name it and includes it as a reference.

RESPONSE: Reference was added.

  1. The distribution of paragraphs is not consistent. (Line 218-221). Does this paragraph only consist of 1 sentence? Please merge with other paragraphs.

RESPONSE: Paragraphs were re-organized as recommended.

  1. Kindly include the last 5 years' references to show that your manuscript is up-to-date.

RESPONSE: Recent references were added.

The manuscript underwent English editing by MPDI English editing service. Certificate was sent to the Editor.

Dear reviewer,

Thank you for your time and thorough report. We do hope that after addressing all your valuable comments you will find that the manuscript has improved.

The manuscript entitled “Fluorescent marker as a tool to promote strategies for environ- mental infection control in dental clinics during and beyond the COVID-19 pandemic” is interesting. However, authors shall revise the following comments before could further consider for publication. Also, the document shall be sent for proofread.

  1. Abstract- Line 21-23. Kindly revise the sentence structure and grammar in this sentence.

RESPONSE: Sentence was revised.

  1. Line 30- Citation needed for this claim “the worldwide state of emergency declared in March 2020 and the rapid spread of the COVID-19 pandemic raised an urgent need to develop simple measures to prevent cross-contamination in medical settings, as well as in other public areas.”.

RESPONSE: Citation was added as requested.

  1. Line 33- Lump references shall be avoided. A maximum of 3 references are recommended for each statement.

RESPONSE: As suggested, some of the references were omitted.

  1. Line 40- “intraoral rotating instruments which disperse microscopic airborne particles posed an increased threat to both dental personnel and patients”. As far as the reviewer’s concerned, saliva is mainly related to direct surface contamination instead of airborne (due to the density and size of the droplets). Unless it is inhale/ exhale process.Please provide some literature to support your claim.

RESPONSE: Thank you for this comment. The Introduction section was substantially expanded to cover this issue.

  1. Line 58-60- The information on surface cleanliness in healthcare-related facilities is very limited. Only one example (fluorescent gel) was mentioned. Sodium hypochlorite solution, an EPA-approved surface disinfectant for healthcare facilities shall be included to improve the content of the present manuscript. Such information is widely reported. Example: https://doi.org/10.1007/s11356-021-16171-9

RESPONSE: Information as well as the reference were added to the text.

  1. Line 74-“The aim of the present study was twofold”. Please revise this sentence. Fragmented sentence.

RESPONSE: The sentence was revised.

  1. Authors shall highlight the discrepancy of past works, and then highlight the gap that this manuscript has addressed.

RESPONSE: New references were added.

  1. Pictures 1-3 shall be labelled. For example in Fig. 3, sink, keyboard, etc. Also, why start with picture instead of figure? Usually, a manuscript only contains of figures, tables, or supplementary documents. Less likely to use pictures as the caption. PLease check the author's guideline of IJERPH

RESPONSE: Labels were added to the figures.

  1. Line 149- Past study used similar samplings no. for pre- and post-examination. Authors shall justify why this study considered a different number of pre-and post-examinations.

RESPONSE: Explanation was added to the text:

As examinations were carried out during the first year of the COVID-19 pandemic, clinical work was restricted by the Ministry of Health, State of Israel (e.g., working periods, the number of active dental units, etc.). Therefore, the number of examined surfaces was different between the two examination periods (574 pre-I and 662 post-I).

  1. Plot of SD on each bar shall be included in Figure 1

RESPONSE: Numbers indicating SD values were added at the top of the bars.

  1. Correlation of surface cleanliness- students cubicles vs. school’s public areas shall be included in Figure 2.

RESPONSE: The figure presents results of a ANOVA with repeated measures: Within-Subjects Factors Public areas vs. Students’ cubicle; between subjects factor Pre I vs. Post I. As the only main effect was between Pre I and Post I findings only these results are quoted in the text (ANOVA, Mean square = 2645.223, F (1) = 10.89, p<0.005).

  1. Discussion- Line 179- which “Guideline” that authors referred to? Kindly name it and includes it as a reference.

RESPONSE: Reference was added.

  1. The distribution of paragraphs is not consistent. (Line 218-221). Does this paragraph only consist of 1 sentence? Please merge with other paragraphs.

RESPONSE: Paragraphs were re-organized as recommended.

  1. Kindly include the last 5 years' references to show that your manuscript is up-to-date.

RESPONSE: Recent references were added.

The manuscript underwent English editing by MPDI English editing service. Certificate was sent to the Editor.

v

Reviewer 3 Report

The paper “Fluorescent marker as a tool to promote strategies for environmental infection control in dental clinics during and beyond the COVID-19 pandemic” by E. Dolev et al. proposes a study to assess and enhance surface cleanliness in the dental clinics and public areas of a major dental school.

A significant improvement in surfaces’ cleanliness was observed after an educational session used to stress the importance of cross-contamination prevention.

The research presented requires some improvements to demonstrate that the results obtained are related to the pretended objectives.

My main concerns about the text are:

Title: The study does not focus on “to promote strategies for environmental infection control” but only on the control of contaminated surfaces and cross-contamination. Please change the title to emphasize this aspect.

Line 44: “SARS-CoV-2 virus … can survive up to several days on surfaces”. Although this is true, it is not so obvious its capacity for contagion from these sources. A depth discussion is required, and it must be included more relevant bibliographic sources, not only one.

Lines 58-60: Fluorescent markers have been employed for other uses, and they must be cited (for example, for dermal evaluation of chemical exposition from the environment).

Line 84, 2.1. Study design. There is no mention of using personal protective equipment (PPE). In this case, the use of gloves, disposable or not. It is a key aspect of the study because if some personnel use disposable gloves, the fluorescent marker (FM) will disappear (for example, the students). But if other personnel use permanent gloves, the FM will be accumulated during the day (for example, cleaning personnel). In this case, the results probably are not related only to the students’ cleaning responsibility. The authors must include this factor in their discussion of the results.

Line 94: “An updated cleaning protocol … was formulated and presented”. I also have some doubts about that step. The use of a disinfectant solution probably deactivates the viability of the virus to infect. However, it is not so evident that the FM disappears from the hands, and its spread to the surfaces will continue. Therefore, a “false positive” is being measured. The authors must also include this factor in their discussion of the results.

Line 102. What is the FM substance?

Table I – Figure I. The label “Students’ cubicles” in Table I is changed to “Dental units” in Figure I. Please, unify this label. In addition, although it is evident, explain in the caption for Figure I the meaning of the asterisk (*).

Author Response

Dear reviewer,

Thank you for your time and thorough report. We do hope that after addressing all your valuable comments you will find that the manuscript has improved.

The paper “Fluorescent marker as a tool to promote strategies for environmental infection control in dental clinics during and beyond the COVID-19 pandemic” by E. Dolev et al. proposes a study to assess and enhance surface cleanliness in the dental clinics and public areas of a major dental school.

A significant improvement in surfaces’ cleanliness was observed after an educational session used to stress the importance of cross-contamination prevention.

The research presented requires some improvements to demonstrate that the results obtained are related to the pretended objectives.

My main concerns about the text are:

COMMENT 1: Title: The study does not focus on “to promote strategies for environmental infection control” but only on the control of contaminated surfaces and cross-contamination. Please change the title to emphasize this aspect.

RESPONSE: Title was rephrased.

COMMENT no.2: Line 44: “SARS-CoV-2 virus … can survive up to several days on surfaces”. Although this is true, it is not so obvious its capacity for contagion from these sources. A depth discussion is required, and it must be included more relevant bibliographic sources, not only one.

RESPONSE:  Thank you for this comment. The subject was expanded, including the addition of references.

COMMENT no.3 : Lines 58-60: Fluorescent markers have been employed for other uses, and they must be cited (for example, for dermal evaluation of chemical exposition from the environment).

RESPONSE: Other use of FMs were quoted in the introduction section and references were added accordinguly.

Line 84, 2.1. Study design. There is no mention of using personal protective equipment (PPE). In this case, the use of gloves, disposable or not. It is a key aspect of the study because if some personnel use disposable gloves, the fluorescent marker (FM) will disappear (for example, the students). But if other personnel use permanent gloves, the FM will accumulate during the day (for example, cleaning personnel). In this case, the results probably are not related only to the students’ cleaning responsibility. The authors must include this factor in their discussion of the results.

RESPONSE: Thank you for this comment. Information regarding the use of PPE was added to the text in the Material and method section 2.1: Study design. The possible differences in the use of gloves between students and cleaning personnel were discussed in the discussion section.

Line 94: “An updated cleaning protocol … was formulated and presented”. I also have some doubts about that step. The use of a disinfectant solution probably deactivates the viability of the virus to infect. However, it is not so evident that the FM disappears from the hands, and its spread to the surfaces will continue. Therefore, a “false positive” is being measured. The authors must also include this factor in their discussion of the results.

RESPONSE: The issue of possible “false positive” was added to the text.

Line 102. What is the FM substance?

RESPONSE: Information about the FM substance used was added to the text in Material and methods section 2.2 evaluation of surfaces.

Table I – Figure I. The label “Students’ cubicles” in Table I is changed to “Dental units” in Figure I. Please, unify this label. In addition, although it is evident, explain in the caption for Figure I the meaning of the asterisk (*).

RESPONSE:  Corrected and explained.

Round 2

Reviewer 2 Report

The content of the manuscript is better after extensive revision.

Reviewer 3 Report

I congratulate the authors for the changes made to the original manuscript. The article has improved the presentation and explanation of the results obtained.